# Tuning Ising superconductivity with layer and spin–orbit coupling in two-dimensional transition-metal dichalcogenides

Sergio C. de la Barrera [1], Michael R. Sinko[1], Devashish P. Gopalan[1], Nikhil Sivadas[1,2], Kyle L. Seyler[3], Kenji Watanabe [4], Takashi Taniguchi[4], Adam W. Tsen[5], Xiaodong Xu [3,6], Di Xiao[1] & Benjamin M. Hunt[1]

Systems simultaneously exhibiting superconductivity and spin–orbit coupling are predicted to provide a route toward topological superconductivity and unconventional electron pairing, driving significant contemporary interest in these materials. Monolayer transition-metal dichalcogenide (TMD) superconductors in particular lack inversion symmetry, yielding an antisymmetric form of spin–orbit coupling that admits both spin-singlet and spin-triplet components of the superconducting wavefunction. Here, we present an experimental and theoretical study of two intrinsic TMD superconductors with large spin–orbit coupling in the atomic layer limit, metallic $2H\text{-}TaS_2$ and $2H\text{-}NbSe_2$. We investigate the superconducting properties as the material is reduced to monolayer thickness and show that high-field measurements point to the largest upper critical field thus reported for an intrinsic TMD superconductor. In few-layer samples, we find the enhancement of the upper critical field is sustained by the dominance of spin–orbit coupling over weak interlayer coupling, providing additional candidate systems for supporting unconventional superconducting states in two dimensions.

[1] Department of Physics, Carnegie Mellon University, Pittsburgh, PA 15213, USA. [2] School of Applied and Engineering Physics, Cornell University, Ithaca, NY 14853, USA. [3] Department of Physics, University of Washington, Seattle, WA 98195, USA. [4] Advanced Materials Laboratory, National Institute for Materials Science, Tsukuba, Ibaraki 305-0044, Japan. [5] Institute for Quantum Computing and Department of Chemistry, University of Waterloo, Waterloo, ON N2L 3G1, Canada. [6] Department of Materials Science and Engineering, University of Washington, Seattle, WA 98195, USA. Correspondence and requests for materials should be addressed to S.C.d.l.B. (email: sergio@phys.cmu.edu) or to B.M.H. (email: bmhunt@andrew.cmu.edu)

Cooper pairing in type-II superconductors is typically destroyed by external magnetic fields due to coupling between the applied field and electron orbital and spin degrees of freedom. For fields applied in the plane of sufficiently thin superconductors, orbital effects are suppressed, providing some protection for superconductivity at enhanced fields. In this limit, the dominant mechanism for breaking superconducting order is Pauli paramagnetism, in which the upper critical field $H_{c2}^{\parallel}$ is given by the Chandrasekhar-Clogston (or Pauli) paramagnetic limit, $H_P \equiv (1.86\,\mathrm{T\,K^{-1}})\,T_{c0}$ at $T = 0$, with superconducting transition temperature, $T_{c0}$[1,2]. However, recent measurements have shown that superconductivity in some atomically thin TMDs survives in the presence of in-plane fields significantly beyond the Pauli limit[3–5]. This effect is proposed to result from a mechanism known as Ising pairing, in which a particular type of Dresselhaus spin–orbit coupling (SOC), termed Ising SOC, pins the electron spins to the out-of-plane direction[6,7], reducing the pair-breaking effect of the in-plane field.

In crystals that lack a center of inversion, symmetry allows for an antisymmetric form of SOC[8]. The 1H-phase of monolayer TMDs is a special case, with both out-of-plane mirror symmetry and broken inversion symmetry (Fig. 1a), restricting the crystal electric field **E** to point in-plane (Fig. 1b). Thus, for electron motion in the same $x$–$y$ plane, antisymmetric SOC gives rise to an effective magnetic field $\mathbf{B}_{so} \propto \mathbf{E} \times \mathbf{k}$ that is directed out-of-plane, leading to a momentum-dependent energy splitting between the spin states $g\mu_B B_{so}(\mathbf{k})$ that changes sign upon inversion through the Brillouin zone center. This spin splitting naturally leads to Cooper pairing between an electron in one of the two spin-split Fermi surfaces around K (the K valley) with its time-reversed pair, of opposite spin and momentum, around K′ (Fig. 1b). The two Fermi surfaces give rise to two distinct populations of Cooper pairs, one each from the upper and lower spin-split bands (Fig. 1c–f), with differing densities of states at the Fermi level.

This basic picture of Ising superconductivity in monolayer TMDs, in which electrons with opposite out-of-plane spins in opposite K and K′ valleys form singlet Cooper pairs, can be complicated by many other effects: additional Cooper pair channels allowed by the band structure beyond K and K′ pairing, coupling between the layers in few-layer samples, the presence of spin-triplet Cooper pairing, guaranteed by strong antisymmetric SOC[8], and extrinsic effects such as spin–orbit scattering (SOS)[9] and intervalley scattering[10]. The relative importance of these effects in modifying the Ising protection of $H_{c2}^{\parallel}$ is an open experimental and theoretical question.

In this work, we study $2H_a$-TaS$_2$, an intrinsic TMD superconductor with the same crystal symmetry and similar electronic structure as NbSe$_2$, but with stronger SOC. Experimentally, we compare the superconducting properties of atomically thin $2H_a$-TaS$_2$, with a large atomic SOC contribution from the heavy Ta atoms, with those of $2H_a$-NbSe$_2$ (hereafter TaS$_2$ and NbSe$_2$, respectively). We isolate ultrathin TaS$_2$ to the monolayer limit, confirming for the first time that there is in fact a stable 1H polytype with a superconducting phase, and extend existing measurements[11] of $T_{c0}$ as a function of the number of layers $N$ down to the monolayer limit. We show that the upper critical field $H_{c2}^{\parallel}(T)$ is significantly enhanced in monolayer TaS$_2$ relative to NbSe$_2$, compelling evidence of the Ising SOC origin of pairing protection in these intrinsic metallic TMDs. We perform first-principles calculations of the band structures and Fermi surfaces of monolayer TaS$_2$ and NbSe$_2$, including spin–orbit coupling, and we analyze the bands to quantify the role of additional Cooper pairing in the Γ pocket of the Fermi surface. We measure $H_{c2}^{\parallel}(T)$ in several few-layer devices of TaS$_2$ and NbSe$_2$ and observe a large enhancement of $H_{c2}^{\parallel}$ above $H_P$ in 2L and 4L devices which is close to that of 3L and 5L devices, despite the restoration of inversion

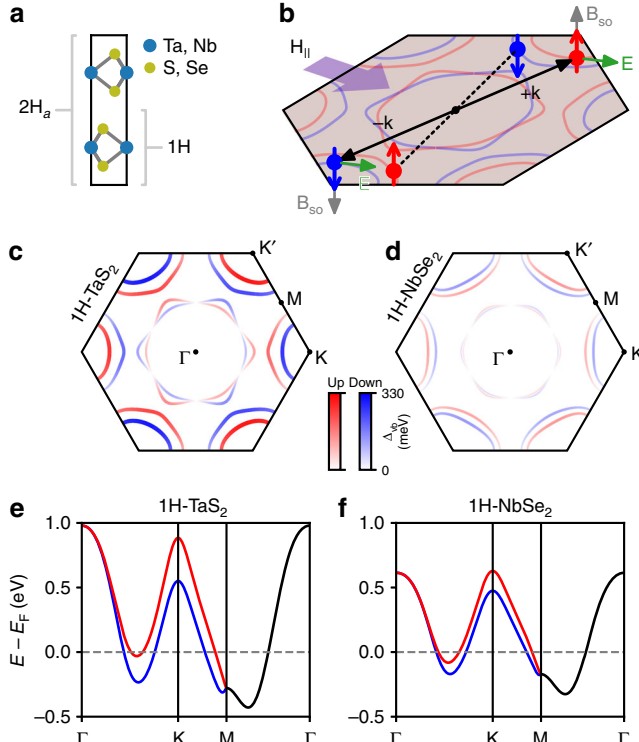

**Fig. 1** Electronic structure of monolayer metallic transition-metal dichalcogenides. **a** Crystal structure of $2H_a$-MX$_2$ (with transition metal atoms directly above one another along the $c$-axis), viewed along [100] direction for M $\in$ {Nb, Ta} and X $\in$ {S, Se} with 1H (monolayer) substructure indicated. **b** Electrons in the K and K′ valleys with spins pinned to the out-of-plane direction due to effective field $\mathbf{B}_{so} \propto \mathbf{E} \times \mathbf{k}$ resulting from planar crystal field and momentum. Straight black lines connect time-reverse pairs. **c** Spin-projected Fermi surface of monolayer TaS$_2$ and **d** NbSe$_2$ computed by density functional theory (DFT). Red corresponds to one $S_z$ projection and blue to the opposite (e.g., up and down, respectively). Variation in the shading and curve thickness indicates the magnitude of spin-splitting in the valence band $\Delta_{vb}(\mathbf{k})$ due to spin–orbit coupling, with the color scale being shared between **c** and **d** to emphasize the difference in magnitudes. **e** Relevant bands around the Fermi level for monolayer TaS$_2$ and **f** NbSe$_2$ from DFT, with spin polarization corresponding to colors in **c**, black bands being spin degenerate

symmetry in the even-layer-number devices. To provide insight into this persistent enhancement of $H_{c2}^{\parallel}$, we calculate the interlayer coupling energy $t_{\perp}$ for NbSe$_2$ and TaS$_2$ and show that the trend of $H_{c2}^{\parallel}$ as a function of the number of layers, $N$, depends on the ratio of the interlayer coupling energy and SOC strength $t_{\perp}(N)/\Delta_{so}$. Finally, we measure $H_{c2}^{\parallel}(T)$ in few-layer NbSe$_2$ and TaS$_2$ in a crucial low-temperature regime, down to 300 mK, where differences among the various theoretical models become evident[3–5,10,12,13].

## Results

**Transport in zero field.** We fabricated several multiterminal transport devices from TaS$_2$ and NbSe$_2$ exfoliated from bulk crystals, capped with boron nitride (BN) inside a nitrogen-filled glove box, and contacted with graphite in series with Cr/Pd/Au leads (more details are in the Methods section). Figure 2 shows a measurement of the longitudinal resistance $R_{xx}(T)$ of five samples in our study: bilayer (2L) and trilayer (3L) NbSe$_2$ and of monolayer (1L), trilayer (3L), and five-layer (5L) TaS$_2$. All samples show a transition from the normal state (with resistance $R_n$ of the order of 100 Ω per square for all samples) to a zero-resistance

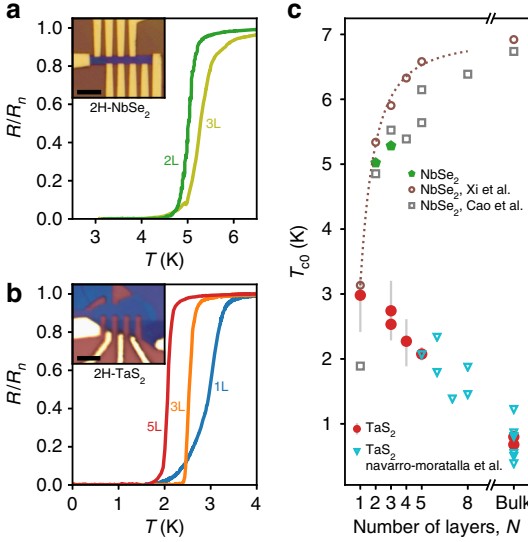

**Fig. 2** Superconductivity of atomically thin TaS$_2$ and NbSe$_2$ in zero magnetic field. **a** Temperature dependence of the normalized longitudinal resistance for bilayer (2L) and trilayer (3L) NbSe$_2$ around the superconducting transition. Optical image of the 3L sample shown in inset. **b** Resistance of monolayer (1L), trilayer (3L), and five-layer (5L) TaS$_2$ samples, exhibiting a reverse trend in the $T_{c0}$ with decreasing thickness. Optical image of 3L sample shown in inset. Scale bars are 4 μm. **c** Compilation of superconducting transition temperatures $T_{c0}$ as a function of thickness for TaS$_2$ and NbSe$_2$ samples from this work as well as from Navarro-Moratalla et al.[11], Xi et al.[4], and Cao et al.[16], using 50% of normal state resistance to define the transitions. Error bars on TaS$_2$ data denote temperature at 10 and 90% of normal state resistances. Dotted line follows the fitting curve used in ref. [4] for NbSe$_2$

state at a temperature $T_{c0}$, which we take by convention to be defined by $R(T_{c0}) = 0.5R_n$. For $T > T_{c0}$, a rounding of the transition is observed that is similar in all of our samples. This is indicative of the enhanced fluctuations in two dimensions and can be described by fitting, for example, the Aslamazov–Larkin or the Halperin–Nelson formulae to these $R(T)$ curves for $T > T_{c0}$[14]. For $T < T_{c0}$, a finite-resistance tail develops with a degree of rounding that varies from sample to sample. In Fig. 2b this is seen clearly if one compares the 1L and 3L data. We ascribe this behavior to effects of the finite size of our samples[14].

TaS$_2$ is known to exhibit a surprising trend in the superconducting critical temperature $T_c$ as a function of thickness[11]; whereas $T_{c0}$ decreases as the number of layers is reduced for NbSe$_2$, in TaS$_2$ the opposite trend is observed down to five layers, the thinnest sample previously reported. Here, we show that this striking trend continues to the monolayer limit (Fig. 2c), however, the mechanism behind this enhancement of $T_{c0}$ is a subject of ongoing debate[15].

For a given layer number $N$, there is significant amount of scatter in the $T_{c0}$ data for both TaS$_2$ and NbSe$_2$. For example, for bilayer NbSe$_2$, measurements of $T_{c0}$ from this work and from refs. [4,16] span the range from 4.9 to 5.3 K. This variation within a given $N$ may be due to effects from the substrate or to varying amounts of disorder from sample to sample. An effect that can have a much larger impact, particularly in the case of TaS$_2$, is intercalation by organic and non-organic molecules[17]. To exclude the possibility of unintentional intercalation of the TaS$_2$ crystals, we performed control experiments on bulk devices fabricated alongside the 1L and 3L samples, subject to the same fabrication processes (see Supplementary Note 3 for details).

To obtain the cleanest results possible, the data shown in Fig. 2 were taken within a few days of exfoliation of each crystal.

However, despite the h-BN encapsulation intended to protect the TaS$_2$ crystals during the brief periods of ambient exposure between experiments, we did observe noticeable degradation in the superconducting properties within a few weeks to a few months of the devices being fabricated for all of the devices. For the 1L device in particular, we found that the monolayer portion of the device degraded away entirely over a period of 2 months, leaving open the possibility that an even cleaner 1L sample might exhibit even more pronounced enhancement than what we report here.

**Magnetotransport in parallel and perpendicular fields**. We turn now to our investigation of atomically thin TMD superconductors in the presence of magnetic fields perpendicular and parallel to the 2D plane. Figure 3 shows the behavior of representative devices of TaS$_2$ and NbSe$_2$. In perpendicular field, superconductivity is destroyed when the total area occupied by the normal cores of vortices is comparable to the total area of the sample, as in three-dimensional (type-II) superconductors. This leads to the Ginzburg–Landau expression for the upper critical field, $H_{c2}^{\perp}(T) = \frac{\Phi_0}{2\pi\xi_{GL}^2(0)}\left(1 - T/T_{c0}\right)$, which allows us to estimate the coherence length $\xi_{GL}(0) \approx 20$ nm for the 3L TaS$_2$ and $\approx 10$ nm for the 2L NbSe$_2$. In both of these samples, and indeed in all of the devices that we have studied, at finite perpendicular fields less than $H_{c2}^{\perp}$, the resistance of the devices does not go to zero as $T \to 0$ but rather saturates to a finite value (Fig. 3a, c). The nature of this finite-conductivity state at $T = 0$ and $H_\perp \neq 0$ has been discussed in refs. [18,19] and further discussion will be deferred to a future work, but we note it here to distinguish the zero-temperature behavior in perpendicular field from that in parallel field.

In Fig. 3b, d, we show the dependence of the resistance of the same TaS$_2$ and NbSe$_2$ devices as the parallel magnetic field (in the plane of the 2D crystals) is varied at fixed temperatures. For some devices we also perform the measurement of $H_{c2}^{\parallel}$ by fixing the parallel field and sweeping the temperature (Methods). At the lowest temperatures, superconductivity in the atomically thin crystals survives up to very large parallel magnetic fields: 25 T for 3L TaS$_2$ and 28 T for 2L NbSe$_2$, corresponding to an anisotropic enhancement $H_{c2}^{\parallel}/H_{c2}^{\perp}$ of 32× and 8× the upper critical fields in the perpendicular orientation, respectively. The anisotropy is even larger for monolayer TMDs, as will be discussed next.

In monolayer TaS$_2$, we find that for $T < 2$ K, the upper critical field in the parallel orientation is larger than the highest field available (34.5 T; Fig. 4a) in the experimental apparatus, whereas in a perpendicular field the superconductivity is quenched at a field near 1.2 T as $T \to 0$. The qualitative behavior of monolayer NbSe$_2$ is similar[4], but with slightly modified temperature and field scales. To facilitate a quantitative comparison between the two materials, we plot the in-plane upper critical field $H_{c2}^{\parallel}(T)$ normalized to the Pauli limit $H_p$ vs. the reduced temperature $T/T_{c0}$. Figure 4 shows a summary of our $H_{c2}^{\parallel}(T)$ data for 1L, 3L, 4L, and 5L TaS$_2$ (Fig. 4a), along with 2L NbSe$_2$ superimposed with 1L-NbSe$_2$ data (Fig. 4b) from ref. [4]. On this scale, it is clear that these materials continue to superconduct well above the Pauli limit, $H_p$, and that the slope of the phase boundary $dH_{c2}^{\parallel}/dT$ between normal and superconducting states near $T/T_{c0} \to 1$ is strikingly steeper for the monolayer samples compared to the few-layer ones.

## Discussion

In the absence of orbital effects, the upper critical field $H_{c2}(T)$ is determined by comparing the superconducting condensation energy with the spin paramagnetic energy in the presence of an external field $H$. Allowing for a finite spin susceptibility $\chi_s$ in the superconducting state, when $H = H_{c2}(0)$ the energy balance

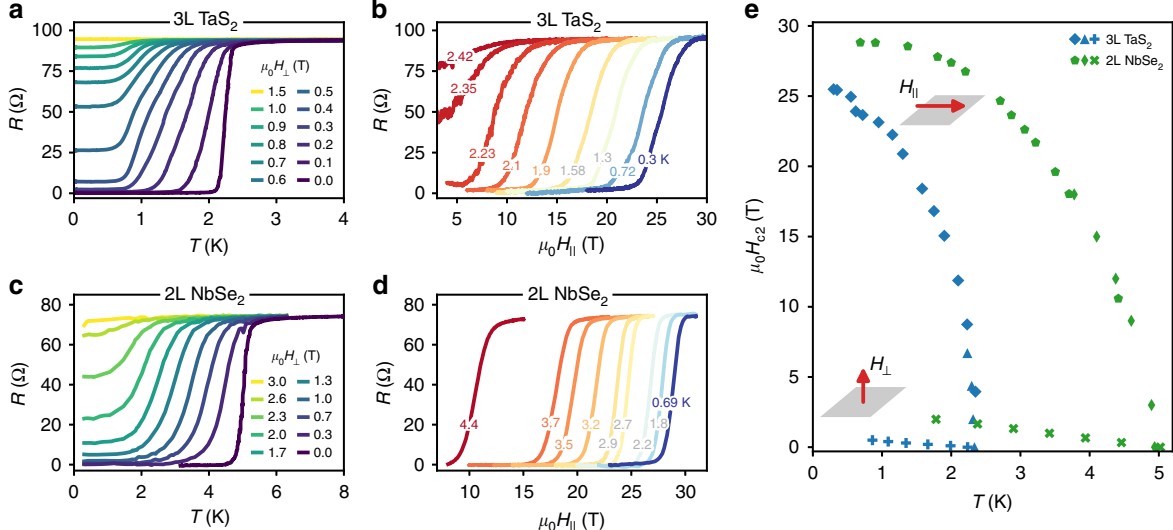

**Fig. 3** Perpendicular and parallel magnetic field dependence. **a** Temperature dependence of longitudinal resistance of trilayer (3L) $TaS_2$ in the presence of an applied magnetic field $H_\perp$ in the out-of-plane direction. **b** Magnetic field dependence of the $TaS_2$ resistance for a field $H_\parallel$ applied in an in-plane direction, for a few constant temperatures as indicated. The field value $H_{c2}^\parallel$ at which the resistance transitions to a zero-resistance state at a fixed temperature is equivalent to the transition temperature $T_c$ of the superconducting state for a fixed field. **c** Temperature dependence of the bilayer (2L) $NbSe_2$ sample for a few perpendicular fields. **d** In-plane field dependence of the same $NbSe_2$ sample for a range of constant temperatures. **e** Temperature dependence of the upper critical field $H_{c2}$ of both samples as extracted from the 50% normal state resistances from the data shown in **a**–**d**

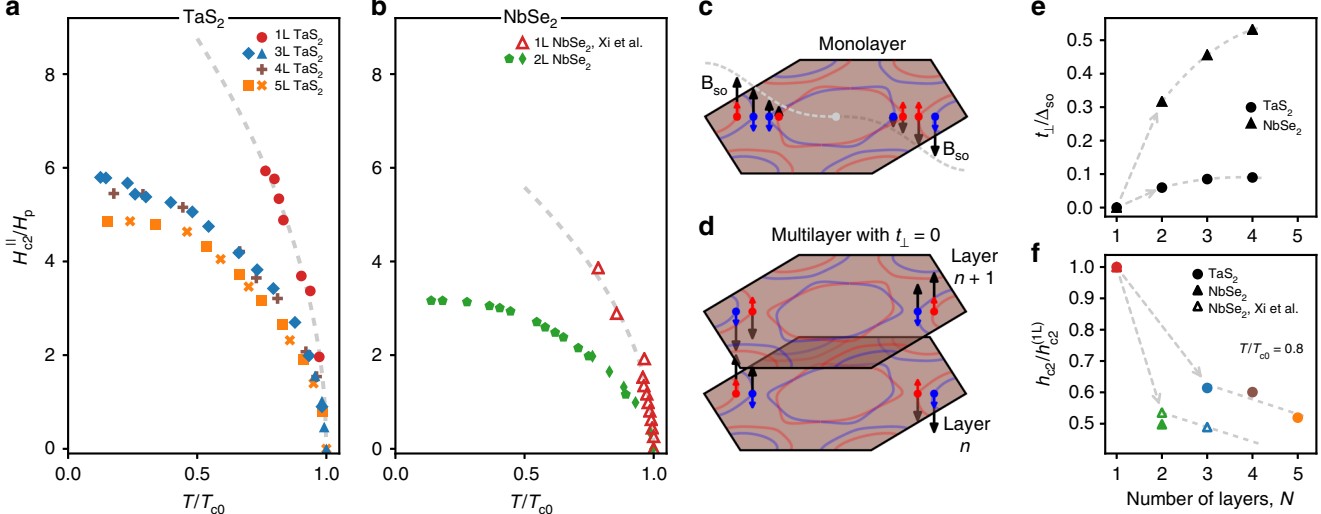

**Fig. 4** Ising superconductivity in single-layer and few-layer $TaS_2$ and $NbSe_2$. **a** Parallel upper critical field normalized to Pauli paramagnetic limit, $H_{c2}^\parallel/H_p$, as a function of reduced temperature $T/T_{c0}$ for monolayer (1L), trilayer (3L), four-layer (4L), and five-layer (5L) $TaS_2$ samples, and **b** monolayer (1L, from Xi et al.[4]) and bilayer (2L) $NbSe_2$. Dashed lines show the square-root fit described in the main text. **c** Illustration of spin states along intersection Fermi surface and line of high symmetry from 1L $TaS_2$, experiencing varying $\mathbf{B}_{so}(\mathbf{k})$ throughout Brillouin zone. Spin projections indicated by color. Dashed line marks the envelope of $\mathbf{B}_{so}$ along K'–Γ–K. **d** Schematic Fermi surface of multilayer $TaS_2$ ignoring interlayer coupling, $t_\perp$, showing opposite $\mathbf{B}_{so}$ in neighboring layers due to inversion of crystal field between layers. **e** Ratio of interlayer coupling energies to spin–orbit coupling strength $t_\perp/\Delta_{so}$ as a function of the number of layers for $TaS_2$ and $NbSe_2$ extracted from density functional theory, with $\Delta_{so}^{TaS_2} = 122$ meV and $\Delta_{so}^{NbSe_2} = 49.8$ meV. Dashed lines are provided as guides to the eye. **f** Dependence of the reduced upper critical field $h_{c2} \equiv H_{c2}^\parallel/H_p$ evaluated at $T/T_{c0} = 0.8$ on the number of $TaS_2$ or $NbSe_2$ layers, including $NbSe_2$ data from Xi et al.[4]. Data are normalized by the reduced upper critical field of a monolayer $h_{c2}^{(1L)}$ to enable direct comparison

$\frac{1}{2}N(E_F)\Delta_0^2 + \frac{1}{2}\chi_s H^2 = \frac{1}{2}\chi_n H^2$[20] leads to

$$H_{c2}(0) = \sqrt{\frac{N(E_F)\Delta_0^2}{\chi_n - \chi_s}}, \quad (1)$$

with density of states at the Fermi level $N(E_F)$, BCS gap $\Delta_0$, and susceptibilities in the normal and superconducting states, $\chi_n$, $\chi_s$. For a spin-singlet superconductor without spin–orbit coupling,

the spin susceptibility in the superconducting state $\chi_s$ goes to zero as $T \to 0$, whereas in the normal state the susceptibility is given by the Pauli paramagnetic susceptibility $\chi_n = \chi_p \equiv 2\mu_B^2 N(E_F)$. Together, the difference in susceptibilities yields the Pauli limiting field, $H_p = \Delta_0/\sqrt{2}\mu_B$. For a weak-coupling BCS gap, $\Delta_0 \equiv 1.76k_B T_{c0}$, and thus $H_p = (1.86 \text{ T K}^{-1})T_{c0}$.

As shown in Fig. 4a, b for in-plane fields, the trend toward $H_{c2}^\parallel(0)$ is many times larger than the Pauli limit $H_p$ for all

samples. To explore the origin of this enhancement, we consider three important effects of strong Ising SOC ($\Delta_{so} \gg \Delta_0$) that can lead to $H_{c2}^{\parallel} > H_p$: modification of the normal state susceptibility $\chi_n < \chi_p$, enhancement of the spin susceptibility in the superconducting state from both spin-singlet and spin-triplet Cooper pairing, and, in the presence of disorder, the possibility of SOS.

One possible source of $H_{c2}^{\parallel} > H_p$ enhancement in TaS$_2$ and NbSe$_2$ is the reduced spin susceptibility arising from the competition between the strong Ising spin–orbit field in $\mathbf{k}$-space with the external in-plane field (Fig. 1b), an effect known as van Vleck paramagnetism[4,20,21]. In the superconducting state, the ability of an in-plane field to break up Cooper pairs is weakened relative to the Pauli paramagnetic case, enabling the upper critical field to exceed the Pauli limit. Close to $T_{c0}$, the experimental $H_{c2}^{\parallel}(T)$ approaches a square-root dependence, $H_{c2}^{\parallel}(T) \approx H_0\sqrt{1 - T/T_{c0}}$, familiar from 2D Ginzburg–Landau (GL) theory and also the asymptotic form of several existing theories for $H_{c2}^{\parallel}(T)$ mentioned in this work. To compare the 1L TaS$_2$ data to 1L NbSe$_2$, we fit the parallel field data to the square-root dependence close to $T_{c0}$, with $H_0$ as a free parameter (Fig. 4a, b). We find $H_0 = 65.6$ T for TaS$_2$ and $H_0 = 43.6$ T for NbSe$_2$, yielding a ratio $H_{c2\parallel}^{TaS_2}/H_{c2\parallel}^{NbSe_2} \approx 1.50$. For 1L NbSe$_2$, Xi et al.[4] estimate that $H_0 \approx \sqrt{H_{so}H_p}$ by comparing the superconducting gap to the van Vleck paramagnetic energy, where $H_{so} \sim B_{so}/\mu_0$ is a single number proportional to the strength of the Ising spin–orbit field $\mathbf{B}_{so}(\mathbf{k})$ throughout the Brillouin zone. Since 1L TaS$_2$ and 1L NbSe$_2$ share the same $T_{c0} \approx 3$ K and therefore the same $H_p = 5.5$ T, we expect the ratio of their upper critical fields to be equal to the ratio of the square root of their Ising spin–orbit fields. Using density functional theory (DFT) to compute the band structure and Fermi surface of 1L TaS$_2$ and 1L NbSe$_2$ (Fig. 1c–f), we obtain the $\mathbf{k}$-dependent spin–orbit splitting in the valence band, $\Delta_{vb}(\mathbf{k})$. We may therefore estimate the upper critical field ratio by defining a $\mathbf{k}$-space-averaged SOC strength $2\mu_B H_{so} \equiv \Delta_{so} = \langle \Delta_{vb}(\mathbf{k}) \rangle_{\mathbf{k}_F}$, where the right side of the expression denotes the average over the Fermi surface (in one irreducible wedge of the Brillouin zone), including both K and $\Gamma$ pockets. Using this definition, we obtain the Fermi surface averages $\Delta_{so}^{TaS_2} = 122$ meV, $\Delta_{so}^{NbSe_2} = 49.8$ meV and compute $H_{c2\parallel}^{TaS_2}/H_{c2\parallel}^{NbSe_2} = \sqrt{\Delta_{so}^{TaS_2}/\Delta_{so}^{NbSe_2}} = 1.57$, in reasonable agreement with the experimental ratio of 1.50. Experimentally, we see that the relative change of $H_{c2}^{\parallel}$ with SOC goes approximately as the square root of the ratio of SOC-induced band splitting along the Fermi surfaces of TaS$_2$ and NbSe$_2$.

Having established the relative scaling of $H_{c2}^{\parallel}$ with $\Delta_{so}$, we consider the magnitude and temperature dependence of $H_{c2}^{\parallel}(T)$, taking into account the effect of SOC on spin-singlet as well as spin-triplet superconductivity quantitatively. The well-known self-consistent model of Frigeri et al.[22] includes the essential physics of singlet and triplet pairing in clean crystals with broken inversion symmetry and large, antisymmetric SOC. Details of this model and the comparison with experiment are outlined in Supplemental Discussion. We find that employing this model with physical parameters from DFT to compute $H_{c2}^{\parallel}(T)$ tends to vastly overestimate the upper critical line for 1L TaS$_2$. In the singlet-pairing theory, the upper critical field enhancement results from an enhanced susceptibility in the superconducting state, approaching a finite value in the zero-temperature limit, $\chi_s^{\parallel}(T \rightarrow 0) \neq 0$[8]. The prediction for triplet pairing produces an even larger $H_{c2}^{\parallel}$ compared to the pure singlet case, since parallel magnetic fields cannot break the fraction of triplet Cooper pairs with parallel spins, and thus paramagnetic limiting is expected to be entirely absent[22]. However, from the data we observe a clear field

dependence of the critical temperature $T_c$, suggesting the presence of a separate limiting mechanism from the pure triplet case.

One aspect of the Frigeri singlet model that resembles experiment is the trend toward larger $H_{c2}^{\parallel}$ for stronger SOC. A rough estimate of the scaling of the pure singlet model with $\Delta_{so}$ (ignoring the detailed $\mathbf{k}$-dependence) can be estimated for large SOC ($\Delta_{so} \gg \Delta_0$) near $T \approx T_{c0}$, where the gap function reduces to a square-root dependence, $H_{c2}^{\parallel}(T) \approx H_0\sqrt{1 - T/T_{c0}}$ with $H_0 = \Delta_{so}/\left(\mu_B\sqrt{\ln\frac{2\Delta_{so}}{\Delta_0}}\right)$[10]. In this limit we see that the singlet model has a weakly sub-linear dependence on $\Delta_{so}$. Comparing the cases of TaS$_2$ and NbSe$_2$ again using this approximate form of the singlet model, we find that the estimated upper critical field ratio near $T_{c0}$ is $H_{c2\parallel}^{TaS_2}/H_{c2\parallel}^{NbSe_2} = 2.26$ using the same Fermi surface averages for $\Delta_{so}$ computed by DFT as before. Numerically taking the ratio of the full temperature-dependent computed upper critical lines of TaS$_2$ and NbSe$_2$ similarly yields a nearly constant function of temperature with a value close to 2. The empirical value is 1.50, as discussed previously, which aligns with a scaling of $H_{c2}^{\parallel} \propto \sqrt{\Delta_{so}}$, reinforcing the conclusion that the appropriate theoretical treatment must incorporate effects beyond singlet pairing with Ising SOC in order to reproduce the observed tuning of $H_{c2}^{\parallel}$ with $\Delta_{so}$.

Reinforcing this point, computation of the upper critical line for ionic-liquid gated MoS$_2$ has also resulted in an overprediction of the experimental $H_{c2}^{\parallel}(T)$ when only the Ising SOC is included[5]. By incorporating gate-induced Rashba SOC as a competing effect, which tends to tilt the electron spins in-plane, it is possible to achieve partial agreement with experiment using a pure singlet model for strongly gated MoS$_2$ and WS$_2$[3,23]. Significant Rashba SOC is not appropriate for our system, however, and therefore we conclude that another competing mechanism is responsible for the lack of quantitative agreement between experiment and theory.

Since this standard model appears to overpredict $H_{c2}^{\parallel}$ for our samples in the clean limit, we consider the potential influence of disorder. Hall measurements of the 2L NbSe$_2$ device provide an estimate of the mean free path that appears to favor the clean limit, whereas similar measurements of the 5L TaS$_2$ device suggest the presence of disorder (Supplemental Discussion), although we are unable to directly measure the Hall effect in the monolayer system due to constraints of the device fabrication. It may be that our samples are in fact in an intermediate disorder regime, and thus we briefly examine two possible sources of disorder here. Although superconductivity is robust to some forms of disorder (Anderson's theorem), scattering mechanisms which break time-reversal symmetry can contribute to paramagnetic limiting in our samples. Two such mechanisms are SOS and intervalley scattering. In the former case, SOS results in spin-randomization of the scattered quasiparticles, with a SOS time $\tau_{so}$ associated with the average time between resulting spin-flips. By fitting our monolayer $H_{c2}^{\parallel}$ data with a standard SOS model, we extract $\tau_{so} = 9.30$ fs and $\tau_{so} = 22.2$ fs for 1L TaS$_2$ and 1L NbSe$_2$, respectively (Supplemental Discussion for details). These values are lower than, yet comparable to measurements of intercalated bulk TaS$_2$[9] as well as quasi-2D superconducting MoS$_2$[3]. Intervalley scattering (from K to K′, for example) requires a spin-flip and therefore may separately contribute to pair-breaking, thus limiting the upper critical field. Employing the model of Ilić et al.[10], which is an extension of the Frigeri singlet model that includes intervalley scattering, and fitting to our monolayer results, we find that intervalley scattering times need to be on the order of 2 fs for 1L TaS$_2$ and 5 fs for 1L NbSe$_2$ in order to produce a computed upper critical line (using DFT values for $\Delta_{vb}(\mathbf{k})$) close to what is observed in experiment (Supplemental Discussion). These rapid scattering times highlight the magnitude of scattering required to reconcile the disorder-free prediction of $H_{c2}^{\parallel}$ with experiment. As a comparison, for

superconducting $MoS_2$ Ilić et al.[10] found that intervalley scattering times on the order of 2 ps were sufficient to fit the data from ref. [5], three orders of magnitude longer our estimates for $TaS_2$ and $NbSe_2$. From this, we conclude that additional theoretical work is required in order to fully describe our observations of $H_{c2}^{\parallel}$ in $TaS_2$ and $NbSe_2$. Ultimately, such a treatment may simultaneously require a combination of competing effects to provide plausible agreement with experiment.

We now turn our attention to the $H_{c2}^{\parallel}$ data from the few-layer samples. In bilayer crystals the broken inversion symmetry of the monolayer system is restored, with an inversion center appearing between the layers of the bilayer (Fig. 1a). In trilayers, global inversion symmetry is broken again; restored in four-layer crystals, and so on. One might thus expect oscillatory behavior in the strength of Ising superconductivity as a function of the number of layers, however this is not what we observe in few-layer samples. Despite the restored global inversion symmetry in, for example, bilayer $NbSe_2$, the upper critical field remains much higher than the Pauli paramagnetic limit, approaching 29 T (3.5 times $H_p$) as $T \to 0$ (Figs. 3 and 4), though the enhancement of $H_{c2}^{\parallel}$ above $H_p$ is significantly less than for the monolayer samples. The observation of $H_{c2}^{\parallel} > H_p$ also holds for the 3L, 4L, and 5L $TaS_2$ devices as well, with only a weak dependence on the thickness above 1L.

The lack of $H_{c2}^{\parallel}$ oscillation with layer number parity can be understood in terms of weak coupling between the layers. In the limit of zero interlayer coupling, each layer superconducts independently and the strength of Ising SOC is equivalent to the monolayer system (Fig. 4d). With a small amount of tunneling (weak coupling) between $d$ orbitals of Ta atoms in neighboring layers, the single-particle states in each layer will overlap with states experiencing an opposite effective field, $-\mathbf{B}_{so}$, due to the inverted crystal field in the surrounding layers. The net effect is weaker Ising SOC and a reduced degree of upper critical field enhancement compared to the monolayer case, especially for 2L and 3L crystals, with diminishing changes for additional layers beyond that (until the thickness is sufficient to support vortex formation, which destroys superconductivity at a field below the Pauli limit).

To gauge the strength of this effect, we extract an interlayer hopping energy $t_{\perp}$ from DFT-computed bands for 2L, 3L, and 4L $TaS_2$ and $NbSe_2$ (Fig. 4e), excluding SOC (Methods). We estimate $t_{\perp}$ from the average dispersion (splitting) in the out-of-plane direction along the Fermi surface and plot the ratio $t_{\perp}/\Delta_{so}$ in Fig. 4e, defined to be zero for 1L. We find interlayer coupling for states near the Fermi level of $t_{\perp} \sim 10$ meV for $TaS_2$ and ~20 meV for $NbSe_2$, relatively weak compared to the Fermi surface average SOC of $\Delta_{so} = 122$ meV for $TaS_2$ and 49.8 meV for $NbSe_2$. To highlight the trend between increasing $t_{\perp}$ and decreasing $H_{c2}^{\parallel}$ as they vary with layer number $N$, we plot $H_{c2}^{\parallel}(N)$ directly below, in Fig. 4f. We use the reduced quantity $h_{c2} \equiv H_{c2}^{\parallel}/H_p$ evaluated at $T/T_{c0} = 0.8$ for both $TaS_2$ and $NbSe_2$, and we normalize $h_{c2}$ to its value for the 1L device, $h_{c2}^{(1L)}$. For multilayer devices, the value of $h_{c2}/h_{c2}^{(1L)}$ is only weakly dependent on $N$, but diminishes more rapidly for $NbSe_2$ than $TaS_2$ as $N$ is increased, trending inversely with the ratio $t_{\perp}/\Delta_{so}$, which is larger and increases faster for $NbSe_2$ compared to $TaS_2$.

In terms of the underlying crystal symmetries, we interpret the measured weak dependence of $H_{c2}^{\parallel}(N)$ on $N > 1$ to reflect the staggered non-centrosymmetric structure of $2H$-$TaS_2$ and $NbSe_2$, wherein the individual layers lack local inversion symmetry despite globally possessing inversion centers between the layers. This type of structure is also found in the layered superconductor $SrPtAs$, which exhibits a similar enhancement of the paramagnetic limit despite having a global center of inversion[21]. In the case of $SrPtAs$, the individual hexagonal As-Pt superconducting layers can be considered to have locally broken

inversion symmetry and therefore retain some of the physical properties associated with non-centrosymmetric superconductivity, such as enhanced $H_{c2}^{\parallel}$. In contrast, optical second harmonic generation from our samples exhibits the global changes in inversion symmetry for even- and odd-layer samples (Supplementary Note 4), further supporting our conclusion that the monotonic dependence of $H_{c2}^{\parallel}(N)$ results from a mechanism other than global inversion symmetry.

Finally, we comment on the striking difference in $T_{c0}(N)$ between $TaS_2$ and $NbSe_2$. Upon reducing the thickness of $TaS_2$ in the few-layer limit, we find that $T_{c0}$ increases from the bulk value of $\approx 800$ mK up to 3 K in 1L $TaS_2$ (Fig. 2c), however the origin of this enhancement remains to be understood. The reversed trend $\Delta T_{c0}/\Delta N < 0$ observed in ultrathin $TaS_2$ is unusual not only compared to $NbSe_2$, but also in the context of other two-dimensional and layered superconductors (and echoes the well-studied trend of $T_{c0}$ in intercalated bulk $TaS_2$, see Supplementary Note 6). We suggest detailed studies of the layer dependence of potentially competing charge-density-wave (CDW) order as a route to understanding $T_{c0}(N)$ of few-layer $TaS_2$[17,24–26]. In particular, scanning tunneling spectroscopy of the superconducting and CDW gaps in the few-layer limit may also benefit this understanding greatly[15,27,28].

In conclusion, we have shown that encapsulated $TaS_2$ is stable in its monolayer 1H phase and yields the largest in-plane upper critical field $H_{c2}^{\parallel}$ of the superconducting TMD family, without the need for gating. Moreover, the larger $H_{c2}^{\parallel}$ of 1L $TaS_2$ relative to 1L $NbSe_2$ provides strong evidence of the underlying spin–orbit coupling origin of the upper critical field enhancement. For few-layer samples, weak interlayer coupling (or local broken inversion symmetry) leads to a similar enhancement of $H_{c2}^{\parallel}$, potentially extending the useful range of two-dimensional TMD superconductors beyond the monolayers. Two-dimensional TMD superconductors have drawn great interest as potential platforms for hosting exotic states such as topological and modulated superconductivity[7,29–31]. The strength of these effects generally scales with SOC and some of these effects additionally benefit from a large $H_{c2}^{\parallel}$. As such, $TaS_2$ appears to be a leading candidate in the search for these exotic states.

## Methods

**Device fabrication**. We create devices from few-micron-sized flakes of $TaS_2$ and $NbSe_2$ exfoliated from bulk $2H_a$-polytype crystals. The bulk crystals were grown by HQ Graphene. Because both compounds are susceptible to degradation in ambient conditions, we exfoliate the crystals inside of a nitrogen-filled glove box and encapsulate the exfoliated flakes with hexagonal boron nitride (h-BN) in the same environment.

To rule out a crystallographic phase change of the few-layer crystals upon exfoliate from the bulk $2H_a$ form, we consider both the anisotrpy of the upper critical field and the polarization of the second harmonic generation. For example, the enhancement of $H_{c2}^{\parallel}$ in 1L $TaS_2$ over 10× the Pauli field limit relies on the lack on inversion symmetry in the monolayer crystal, ruling out the 1T phase, which is fully centrosymmetric within each layer. Second harmonic generation of all $TaS_2$ devices in the study exhibits a six-fold rose pattern in the azimuthal angle (Supplementary Figure 4a), reflecting the underlying three-fold symmetry of the 1H phase and ruling out monoclinic 1T′ or orthorhombic $T_d$ phases, or a substantial portion of mixed or distorted phases[32].

To make electrical contact to the crystals, we transfer few-layer graphite, which is similarly exfoliated from bulk, and overlap with part of the TMD crystal to create areas with an atomically smooth interface for electrical contact. The overlapping region of graphite/TMD (entirely encapsulated by h-BN on top) is then etched into separate channels to allow four-terminal measurements of the superconducting TMD alone (see Fig. 2 insets). The etched graphite leads extend beyond the h-BN encapsulating layer allowing Cr/Pd/Au leads with top-contact to the graphite to be defined using standard electron-beam lithography techniques.

**Magnetotransport measurements**. Magnetotransport measurements were made using standard low-frequency AC lock-in techniques with SR8x0 series lock-in amplifiers and a Keithley 2400 SourceMeter. The samples were measured in a dilution refrigerator to a minimum temperature of 25 mK and maximum field of 12 T, as well as at the National High Magnetic Field Lab in Tallahassee, Florida in a

He-3 refrigerator to a minimum temperature of 300 mK and a maximum field of 34.5 T.

In parallel field: This measurement is obtained by first fixing the temperature and the magnetic field and then varying the angle between the magnetic field and the sample until a minimum in the resistance is achieved, allowing us to precisely locate the parallel configuration. We then sweep the magnetic field at fixed temperature $T$ and extract $H_{c2}^{\parallel}(T)$ as the value of the field for which $R = R_n/2$, as in Fig. 3e. Measuring $R(T)$ at fixed parallel field and varying temperature is principally equivalent, however operationally we find that it is more difficult to hold a perfect parallel position while also varying temperature, and thus we prefer the consistency of varying the field while holding the temperature fixed. Nevertheless, we do take data in both modes and plot both data sets together, for example, as shown for the 5L $TaS_2$ device in Fig. 4a, with squares coming from field sweeps and crosses coming from temperature sweeps.

**DFT calculations**. The calculations for the first-principles part was performed using the projector augmented wave[33–35] method encoded in Vienna ab initio simulation package (VASP)[35] with the generalized gradient approximation in the parameterization of Perdew, Burke, and Enzerhof[36,37]. An outer shell configuration of $4p^6\,4d^4\,5s^1$, $5d^4\,6s^1$, [Ne] $3s^2\,3p^4$ and [Ar] $4s^2\,4p^4$ were used for Nb, Ta, S, and Se respectively. Structural optimization was performed for monolayers with a vacuum region more than 15. All the ions were relaxed so that the total energies converged to 0.5 meV per atom with a regular $16 \times 16 \times 1$ Monkhorst–Pack grid.

After obtaining the ab initio wave functions from a self-consistent calculation the corresponding Fermi surface was computed utilizing the Wannier interpolation approach[38–40]. When spin–orbit coupling was included, the spin degeneracy of the bands was lifted away from the Γ-point. The Wannier interpolation was performed by projecting onto 22 bands at each $k$-point, 10 from the transition metal $d$ orbitals (spinors) and 12 from the two chalcogen atom $p$ orbitals. The corresponding spin-projection along the Fermi surface was obtained separately from the first-principles calculation using a Monkhorst–Pack grid of $108 \times 108 \times 1$, and was superimposed on the Fermi surface obtained using the Wannier interpolation.

The interlayer coupling in multilayers were obtained by calculating the dispersion (total splitting) of the bands in the out-of-plane direction without including spin–orbit coupling[41]. An interlayer coupling strength $t_\perp$ is then estimated by extracting half of the remaining splitting of the bands near the Fermi level, $t_\perp \equiv \Delta_{vb}(\mathbf{k}_F)/2$ without SOC.

**Data availability**. The data that support the findings of this study are available from the corresponding author upon reasonable request.

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

## Acknowledgements

We acknowledge fabrication assistance from Jingyi Wu and helpful discussions with Y. Yang, K.F. Mak, C.R. Dean, and E. Telford. We also thank D. Graf and E.S. Choi for experimental assistance at the National High Magnetic Field Laboratory. Funding for S. C.d.l.B. was provided by the Charles E. Kaufman Foundation, a supporting organization of The Pittsburgh Foundation, via Young Investigator research grant KA2016-85226.

Work on device fabrication by D.P.G. was supported by the Department of Energy Early Career program under award number DE-SC0018115. Work on device fabrication and measurement by M.R.S. was supported by the National Science Foundation PIRE program under award number 1743717. The high-field measurements were made at the National High Magnetic Field Laboratory. X.X. and D.X. are supported by the Department of Energy, Basic Energy Sciences, Materials Sciences and Engineering Division (DE-SC0012509). NS acknowledges National Science Foundation (Platform for the Accelerated Realization, Analysis, and Discovery of Interface Materials (PARADIM)) under Cooperative Agreement No. DMR-1539918 and Cornell University Center for Advanced Computing for his time at Cornell University.

## Author contributions

S.C.d.l.B., M.R.S., D.P.G., and A.W.T. fabricated the devices. S.C.d.l.B., M.R.S., D.P.G., A. W.T., and B.M.H. performed the magnetotransport experiments. N.S., with supervision from D.X., performed the theoretical calculations. K.W. and T.T. grew the hBN crystals. K.L.S. performed the optical characterization of the devices under the supervision of X.X. S.C.d.l.B. and B.M.H. wrote the paper, with contributions from M.R.S.

## Additional information

**Competing interests:** The authors declare no competing interests.

