## [Peer Review File · Nature Communications]

Reviewers' comments:

Reviewer #1 (Remarks to the Author):

I consider the work suitable for publication after corrections.

This work reports on the dimensionality effects observed in superconducting materials by studying the magneto-transport curves of 2H-TaS₂ and 2H-NbSe₂ with different thicknesses, including the monolayer. 2H-TaS₂ and 2H-NbSe₂. Although the thickness dependence of 2H-NbSe₂ has been already reported by several groups, regarding 2H-TaS₂ the present work confirms down to the monolayer the previous tendency observed by Navarro-Moratalla et al. (Nat. Commun. 2016, 7, 11043), where an enhancement of the superconducting critical temperature (T_c) was observed, in strict contrast with the T_c suppression observed in 2H-NbSe₂.

The authors characterized the atomically-thin layers by magneto-transport curves and second harmonic generation. Remarkably, it must be noted that working with these non-stable monolayers is a challenge that the authors solved by the encapsulation of the TMDCs layers between h-BN. The main interesting point of the article is the analysis of the data based on the Ising superconductor model and the role of the spin-orbit interaction, yielding to large upper critical fields.

The experiments are well designed, the theoretical proposal is interesting, and the manuscript is clear and well-written. Moreover, the topic is timely and interesting. In fact, there is a big controversy regarding which one is the mechanism responsible for the T_c enhancement in 2H-TaS₂ since several different models have been proposed.

I consider it suitable for publication in Nature Communications after addressing the following comments:

1.- The flakes were obtained by the mechanical exfoliation of bulk crystals. However, regarding the starting material:

1.1.- How were the crystals grown? Chemical Vapor Transport (CVT) using iodine?

1.2.- The authors emphasize the nature of the polytype (i.e., the crystal is 2H and not 1T or other) and the intercalation. For that, the authors did second harmonic generation and control experiments. However, it would be interesting to know the composition of the bulk material by simply conventional chemical analysis methods (like elemental analysis or inductively couple plasma mass spectroscopy). It is common to obtain in TMDCs non-stoichiometric compounds like Ta_(1+x)S₂ and that may affect the transport properties. For example, excess of Ta increases the T_c (Ta_{1.05}S₂ has a T_c of 3.5K; J. Chem. Phys. 1975, 62, 967).

1.3.- In the device fabrication (methods, page 6), it is said that "Second harmonic generation of all TaS₂ devices in the study exhibits a six-fold rose pattern in the azimuthal angle, reflecting the underlying three-fold symmetry of the 1H phase and ruling out monoclinic 1T0 and orthorhombic 1T phases.". However, that six-fold rose pattern is not seen neither in the main text nor in the manuscript. It would be interesting to support that sentence with some experimental figure in the supporting information.

2.- The thickness of the flakes was assessed by "transport and AFM" (SI, page 12). It would be of interest to see some AFM image of the devices in order to have an idea about roughness, existence of bubbles or degradation of flakes.

3.- In order to evaluate the sharpness of the transitions to the superconducting state, it would be useful to see the experimental points in the magneto-transport curves (i.e., to see if the sharp

transitions are defined just by two points or there are several points), as it can be observed in Fig S3. In particular, it would be interesting to see the experimental points in the measurements of the three-terminal devices (page 11), in order to have an estimation of the quality of the further background subtraction.

4.- In page 5, the expression "below 1L" in the sentence "...with only a weak dependence on the thickness below 1L", may be interpreted as sub-monolayer (as, for example, it is commonly used in the superconducting films made of metals like Pb or Nb). However, the term sub-monolayer makes no much sense in the context of the 2D materials. The authors may find a better expression.

Reviewer #2 (Remarks to the Author):

The discovery of intrinsic superconductivity in transition-metal dichalcogenide mono- and few-layers of MoS₂ and NbSe₂ attracted a lot of interest recently. The submitted work adds another compound, TaS₂, with similar properties, to this family. Namely, it is found that mono- and few-layers of hBN-capped 1H-TaS₂ are superconducting, and their parallel upper critical field is strongly enhanced above the Pauli limit. This is taken as a signature of Ising pairing. This unconventional superconducting pairing is induced by a large intrinsic spin-orbit coupling specific to the band-structure of these materials. The experimental study on TaS₂ is complemented by results on the already known NbSe₂ compound, which has a similar band-structure. Another important experimental finding is that the enhancement of the upper critical field above the Pauli limit persists in materials with an even number of layers, which is taken as a signature of the weak interlayer coupling, even if the dependence of the critical temperature (at zero field) on the number of layers remains poorly understood. In my opinion, the experimental findings are very important. However, there are inconsistencies in the theoretical analysis provided in the manuscript to analyze the Ising nature of superconductivity. I would suggest the authors to revise the comparison between experiment and theory substantially before I can recommend the publication of the present work in Nature Communications.

Indeed, in agreement with the literature, the authors argue that the electronic bands are qualitatively well described by the Hamiltonian (S1) near K,K' points, and they use DFT to extract the Ising spin-orbit coupling field, which is believed to be the dominant reason for the enhancement above the Pauli limit in this family of materials. However, in agreement with Refs. [3,5], they note that the corresponding prediction for the upper critical line is well above the experimental one. This discrepancy CANNOT be resolved by using the Abrikosov-Gorkov fit given by Eqs. (1) and (S1). Indeed, there is NO theoretical justification for the extraction of a pair-breaking parameter out of Eq. (S3), taking Hamiltonian (S1) as a starting point, as this equation misses the fact that the parallel magnetic field cannot break the fraction of triplet Cooper pairs with parallel spin induced by the Ising spin-orbit field. In particular, the gray dashed line in Figs. 4a, 4b, and S1, and thereby the extraction of the amplitude of the Ising field from the fit, and its comparison with DFT, makes no sense. This is most visible in Fig. S1, where the gray and light-blue lines, plotted for the same value of the spin-orbit parameter, obviously do not match.

Possible reasons for the discrepancy between theory using the extracted parameters from DFT, and experiment, should be discussed, as the value of Rashba coupling tentatively incorporated in Eq. (S6) and Fig. S1 may be unrealistically large. In particular, the authors should consider the following effects:

1) The results from DFT illustrated in Fig. 1 show that considering bands near K,K' points only is probably a too crude approximation. Would the enhancement above the Pauli limit be as pronounced

taking into account multiband effects?

2) The shortness of the coherence length extracted from the temperature dependence of the perpendicular critical field hints towards the diffusive regime. An estimate of the mean free path would be useful to confirm this. And if it is the case, then, it should be clarified if the theoretical analysis shown in Fig. S1, which assumes the clean limit, would be affected.

3) Analysis of the critical line is performed assuming a local, singlet BCS pairing constant. Which other scenarios could be envisioned? The authors should also comment on the perspectives for using TaS₂ for topological superconductivity, given their findings.

Finally, I did not understand what is the difficulty in extracting the interlayer coupling parameter taking into account spin-orbit coupling at the same time.

Reviewer #3 (Remarks to the Author):

This is a high quality paper with solid data obtained from experiments in extreme conditions of low temperature and high magnetic field. The paper discussed the layer dependence of Ising protection in TaS₂ and NbSe₂. I would highly recommend for publication in nature communications. Before accepting this paper, the following crucial points need to be addressed.

1. A general feature of the Ising SC in the valence band is the intrinsic small protection (measured H_{c2}) compared with the large SOC (B_{so} field is very large), which is in sharp contrast with the TMDs showing Ising SC in the conduction band. Namely, although the SOC is very different in these two cases, the level of violation to the Pauli limit is very similar say in the case of MoS₂ and NbSe₂. I think this point need to be address in the first place.

2. I can clearly see the trend from 1 to 5 layer (Fig. 4a). However, what is the general trend from 5L to bulk, I suppose the electronic structure of 5 layer and bulk is not so much different. Therefore, very similar H_{c2} should be measured.

3. related to point 2, since bulk doesn't show superconductivity due to CDW, missing of CDW phase needs discussion. The discussion of Ising without considering CDW need justification. Please note the recent paper about the possibility of polymorphism in Nano Letters, 17, 5567-5571 (2017).

4. From SHG, there is a clear layer dependance for inversion symmetry. From H_{c2} measurement, only monolayer is different from the bilayer and thicker crystals. This at least means there are interlayer effect which reduces H_{c2} . The general trend of the decrease of spin expectation value of TMDc with spin localized within layers, namely hidden spin in some literatures, is that although spin expectation value shows odd-even dependence, there is a global decrease of spin expectation valve with the increase of the number layers. Namely, both SHG and H_{c2} should show oscillatory decrease. I can see this trend in SHG measurement (although bilayer data missing in SHG) but not in H_{c2} measurement. I think it point needs clarification.

5. The normalization of the Δ_{SO} over the gamma point needs justification because the paring is not from gamma point in the model illustrated in the paper.

6. Because the Rashba effect is absent in present samples, the model used in MoS₂ is in general not

applicable for the present case. The comparison in Fig. S1 needs clearer justification.

Firstly, we would like to thank all three referees for their positive and encouraging comments. Based on these comments, we have made substantial revisions to the manuscript, including two new supporting figures in the Supplementary Information and a revised Discussion section. In our view, these changes have resulted in an improved manuscript and a clearer presentation of our results.

Below we address individual concerns and comments of each referee.

Reviewer #1:

We thank Referee #1 for pointing out that “the experiments are well designed, the theoretical proposal is interesting, and the manuscript is clear and well-written. Moreover, the topic is timely and interesting.” To address his/her comments, we have made the following changes.

1.1.- How were the crystals grown? Chemical Vapor Transport (CVT) using iodine?

The growth method is indeed chemical vapor transport, prepared by HQ Graphene; now mentioned explicitly in the Methods on line 467 and the Control Experiments section of the Supplementary Info, beginning on line 807.

1.2.- The authors emphasize the nature of the polytype (i.e., the crystal is 2H and not 1T or other) and the intercalation. For that, the authors did second harmonic generation and control experiments. However, it would be interesting to know the composition of the bulk material by simply conventional chemical analysis methods (like elemental analysis or inductively couple plasma mass spectroscopy). It is common to obtain in TMDCs non-stoichiometric compounds like $Ta_{(1+x)}S_2$ and that may affect the transport properties. For example, excess of Ta increases the T_c ($Ta_{1.05}S_2$ has a T_c of 3.5K; J. Chem. Phys. 1975, 62, 967).

The stoichiometry was determined from energy-dispersive x-ray spectroscopy (EDX) to be 67.2% sulfur and 32.8% tantalum for one bulk crystal and 69.6% sulfur and 35.5% tantalum for another, indicating a slight excess of sulfur. Excess sulfur is not known to affect superconductivity in TaS_2 . For our devices, as pointed out by the reviewer, from transport measurements of T_c of bulk crystals exfoliated simultaneously with our atomically thin crystals, we know that the bulk transitions are as expected for stoichiometric TaS_2 . This point is now addressed and clarified in a re-written paragraph in the Control Experiments section of the Supplementary Info, beginning on line 808. We thank the reviewer for pointing out the additional reference regarding stoichiometry; this reference has been added to the paragraph as well.

1.3.- In the device fabrication (methods, page 6), it is said that “Second harmonic generation of all TaS_2 devices in the study exhibits a six-fold rose pattern in the azimuthal angle, reflecting the underlying three-fold symmetry of the 1H phase and ruling out monoclinic 1T0 and orthorhombic Td phases.”. However, that six-fold rose pattern is not seen neither in the main text nor in the manuscript. It would be interesting to support that sentence with some experimental figure in the supporting information.

We appreciate this suggestion; a representative pattern (from the 1L device) has been added to Fig. S5 and is now referenced in the text.

2.- The thickness of the flakes was assessed by “transport and AFM” (SI, page 12). It would be of interest to see some AFM image of the devices in order to have an idea about roughness, existence of bubbles or degradation of flakes.

A new figure (Fig. S2) has been added to the supplementary material which shows AFM images of the four TaS₂ devices for which magnetotransport data is shown in the main text. In a new section discussing these images, we point out that while we do not find bubbles between our h-BN and superconducting layers, we do see signs of degradation that appear along the edges of etched regions of the superconducting crystals. We also enumerate the RMS roughness of each of our TaS₂ device channels, which range from 0.3 nm to 1.3 nm.

3.- In order to evaluate the sharpness of the transitions to the superconducting state, it would be useful to see the experimental points in the magneto-transport curves (i.e., to see if the sharp transitions are defined just by two points or there are several points), as it can be observed in Fig S3. In particular, it would be interesting to see the experimental points in the measurements of the three-terminal devices (page 11), in order to have an estimation of the quality of the further background subtraction.

There are over 200 points along the sharpest part of each transition for the data in the main text. We have plotted this data using points instead of lines and found that the visual difference is imperceptible. As an example, the number of points shown in Fig. S4, which is lower than all the other data shown in the manuscript (including the three-terminal devices), still produces a high density of points in the transition region.

4.- In page 5, the expression “below 1L” in the sentence “...with only a weak dependence on the thickness below 1L”, may be interpreted as sub-monolayer (as, for example, it is commonly used in the superconducting films made of metals like Pb or Nb). However, the term sub-monolayer makes no much sense in the context of the 2D materials. The authors may find a better expression.

We thank Referee #1 for pointing out this typographical error. The sentence has been corrected to “above 1L.”

Reviewer #2:

We thank Referee #2 for commenting on the “important experimental finding ...that the enhancement of the upper critical field above the Pauli limit persists in materials with an even number of layers, which is taken as a signature of the weak interlayer coupling,” and for adding that “the experimental findings are very important.” We address the concerns of Referee #2 below.

Indeed, in agreement with the literature, the authors argue that the electronic bands are qualitatively well described by the Hamiltonian (S1) near K,K' points, and they use DFT to extract the Ising spin-orbit coupling field, which is believed to be the dominant reason for the enhancement above the Pauli limit in this family of materials. However, in agreement with Refs. [3,5], they note that the corresponding prediction for the upper critical line is well above the experimental one. This discrepancy CANNOT be resolved by using the Abrikosov-Gorkov fit given by Eqs. (1) and (S1). Indeed, there is NO theoretical justification for the extraction of a pair-breaking parameter out of Eq. (S3), taking Hamiltonian (S1) as a starting point, as this equation misses the fact that the parallel magnetic field cannot break the fraction of triplet Cooper pairs with parallel spin induced by the Ising spin-orbit field. In particular, the gray dashed line in Figs. 4a, 4b, and S1, and thereby the extraction of the amplitude of the Ising field from the fit, and its comparison with DFT, makes no sense. This is most visible in Fig. S1, where the gray and light-blue lines, plotted for the same value of the spin-orbit parameter, obviously do not match.

We would like to extend special thanks to Referee #2 for raising this point. We agree with the reviewer regarding the use of the phenomenological pair-breaking equation for analyzing the

experimental upper critical line of monolayer TMDs. After careful thought and consideration, we have substantially revised and expanded the Discussion section, addressing several existing models for the upper critical line in 2D superconductors with strong antisymmetric SOC and broken inversion symmetry. In summary, we considered the roles of spin-singlet and spin-triplet pairing according to the theory of Frigeri *et al.* (Ref. 22) for crystals with strong SOC and broken inversion symmetry. We then discuss the possible role of disorder and comment on the relevance and success of each model regarding our upper critical field measurements and suggest further theoretical investigation to resolve what in the end, is an open scientific question. Details of this new commentary appear in the revised Discussion with further details provided in a new, dedicated section of the Supplementary Information.

Possible reasons for the discrepancy between theory using the extracted parameters from DFT, and experiment, should be discussed, as the value of Rashba coupling tentatively incorporated in Eq. (S6) and Fig. S1 may be unrealistically large. In particular, the authors should consider the following effects:

1) *The results from DFT illustrated in Fig. 1 show that considering bands near K, K' points only is probably a too crude approximation. Would the enhancement above the Pauli limit be as pronounced taking into account multiband effects?*

Although the effect of weaker Ising SOC from the Γ -pocket Cooper pairs is not enough to reconcile the difference between experiment and theory (e.g. from Refs. 3 and 5), it does have a pronounced effect on the predicted upper critical line using DFT parameters. Indeed, including the Γ -pocket reduces the predicted upper critical field by nearly a factor of 2 for most temperatures, although the XXXYYY. To make this point explicit, we now include computed curves (1) including only spin-orbit splitting from the K and K' point and (2) from K, K' , and Γ together (the full Fermi surface). These curves are plotted together with the 1L TaS₂ experimental data in a new supplementary figure, Fig. S1.

2) *The shortness of the coherence length extracted from the temperature dependence of the perpendicular critical field hints towards the diffusive regime. An estimate of the mean free path would be useful to confirm this. And if it is the case, then, it should be clarified if the theoretical analysis shown in Fig. S1, which assumes the clean limit, would be affected.*

The referee brings up an important point which has scarcely been discussed in other papers about Ising superconductivity. In our work, a rough estimate of the Drude mean free path from Hall measurements of our 5L TaS₂ sample hints at the presence of disorder (see revised Discussion beginning on line 322 and new section in Supplementary). Geometric constraints for our other samples preclude obtaining Hall measurements. Nevertheless, given the mismatch between our experimental upper critical fields and the clean singlet theory, we do believe that disorder plays an important role. Thus we now include analysis considering both spin-orbit scattering and intervalley scattering mechanisms. Both are included in the revised Discussion (line 320) and the new section of the Supplementary Info with supporting curves in Fig. S1.

3) *Analysis of the critical line is performed assuming a local, singlet BCS pairing constant. Which other scenarios could be envisioned? The authors should also comment on the perspectives for using TaS₂ for topological superconductivity, given their findings.*

Due to large SOC, in principle we expect mixed parity pairing (singlet + triplet) to contribute. Experimentally, from upper critical field measurements it is not possible to separate the effects of singlet and triplet pairing in our samples. However we now address the expected behavior of the induced fraction of triplet Cooper pairs in the revised Discussion (line 265) with details in a dedicated section of the Supplementary Info (beginning line 730). Beyond this, while it is possible that the superconductivity in our samples could involve exotic pairing such as modulated superconductivity, we do not see any strong indicators in our measurements. However, we would like to highlight the favorable properties of single- and few-layer TaS₂ toward the search for such novel superconducting states, therefore we have added a sentence introducing the topic of topological superconductivity early in the abstract, and also elaborated on the potential advantage of TaS₂ over other 2D superconductors in the search for topological superconductivity and other exotic superconducting states in the concluding paragraph, beginning on line 457.

Finally, I did not understand what is the difficulty in extracting the interlayer coupling parameter taking into account spin-orbit coupling at the same time.

We apologize for being unclear. Our goal is to obtain separate spin-orbit coupling Δ_{so} and interlayer coupling t_{\perp} contributions to the total valence band splitting $\Delta_{\text{vb}} \sim \sqrt{\Delta_{\text{so}}^2 + t_{\perp}^2}$ computed by DFT. To achieve this, we follow the method of Ref. 40: First, we extract the total valence band splitting Δ_{vb} from DFT computations including spin-orbit coupling (SOC), and then we extract the interlayer coupling t_{\perp} from separate computations excluding SOC.

Reviewer #3:

We are grateful to Referee #3 for the positive comments and for stating that “this is a high quality paper with solid data,” and that he/she “would highly recommend for publication in Nature Communications.” We address the additional points raised by Referee #3 below.

1. A general feature of the Ising SC in the valence band is the intrinsic small protection (measured H_{c2}) compared with the large SOC (B_{so} field is very large), which is in sharp contrast with the TMDs showing Ising SC in the conduction band. Namely, although the SOC is very different in these two cases, the level of violation to the Pauli limit is very similar say in the case of MoS₂ and NbSe₂. I think this point need to be address in the first place.

We agree that the effect of SOC is significantly different between MoS₂ and NbSe₂ due to the relatively small splitting in the conduction band (relevant for MoS₂ and WS₂) compared to the valence band (relevant for NbSe₂ and TaS₂). The description of Ising superconductivity in the semiconducting TMDs, however, is likely complicated by the ionic gating methods used to induce superconductivity, the qualitative difference in the conduction band Fermi surface, and the details of the accumulation layer and vertical charge distribution created by strong gating. We have included a discussion of superconductivity in MoS₂ on lines 309–319 to (1) highlight the competing effect of Rashba SOC that was required in MoS₂ in order to find agreement between experiment and theory, and (2) to emphasize that this effect is absent in the metallic TMDs, thus requiring other competing effects such as disorder to model the upper critical field of the latter. This point is reiterated at the end of a new section in the Supplemental Information addressing scattering, beginning on line 770.

2. I can clearly see the trend from 1 to 5 layer (Fig. 4a). However, what is the general trend from 5L to bulk, I suppose the electronic structure of 5 layer and bulk is not so much different. Therefore, very similar H_{c2} should be measured.

As the thickness increases beyond 5 layers, we agree that the electronic structure will not change qualitatively from the few-layer case due to weak interlayer coupling. However, the H_{c2} will be substantially reduced (below the Pauli limit) for thicknesses sufficient to allow vortex formation (an orbital rather than a spin effect). In bulk samples, vortex formation is therefore the dominant mechanism leading to the destruction of superconductivity. See, for example, the in-plane $H_{c2}(T)$ of bulk NbSe₂ in Fig. 4 of Ref. 4. We have modified the sentence on line 389 to clarify this point.

3. Related to point 2, since bulk doesn't show superconductivity due to CDW, missing of CDW phase needs discussion. The discussion of Ising without considering CDW need justification. Please note the recent paper about the possibility of polymorphism in Nano Letters, 17, 5567-5571 (2017).

Bulk TaS₂ exhibits superconductivity near 600 mK to 700 mK, as shown in Fig. S4. However we agree that a possible coexisting CDW phase could play a role in the characteristics of the superconducting state in ultrathin TaS₂. The possible role of a diminishing CDW phase in few-layer TaS₂ has recently been covered in an unpublished work (arXiv: 1711.00079). On the other hand, while CDW may play a role in layer dependence of T_{c0} , it is not obvious that it will have any further effect on H_{c2} (i.e. no field dependence). To shed light on this point, in the conclusion we suggest detailed studies of the layer dependence of potentially competing CDW order with scanning tunneling spectroscopy of the super-conducting and CDW gaps, beginning on line 441. Finally, we thank the reviewer for pointing out the reference pertaining to additional polymorphism in 2D TaS₂. We have added this reference to our Methods section discussing our characterization of the 2H and 1H polytypes (line 484).

4. From SHG, there is a clear layer dependence for inversion symmetry. From H_{c2} measurement, only monolayer is different from the bilayer and thicker crystals. This at least means there are interlayer effect which reduces H_{c2} . The general trend of the decrease of spin expectation value of TMDc with spin localized within layers, namely hidden spin in some literatures, is that although spin expectation value shows odd-even dependence, there is a global decrease of spin expectation value with the increase of the number layers. Namely, both SHG and H_{c2} should show oscillatory decrease. I can see this trend in SHG measurement (although bilayer data missing in SHG) but not in H_{c2} measurement. I think it point needs clarification.

We agree that the lack of oscillatory behavior in H_{c2} with the number of layers, despite global symmetry changes, is an interesting result (line 80). We attribute this finding to weak interlayer coupling in TaS₂ and NbSe₂. This masks the change in global inversion symmetry between even and odd layer thicknesses, enhancing the effect of broken local inversion symmetry instead, like a stack of weakly-coupled monolayers (lines 380—431). To better illustrate this point, we have specifically mentioned the contrasting dependence of SHG on layer number on line 427.

5. The normalization of the Δ_{SO} over the gamma point needs justification because the pairing is not from gamma point in the model illustrated in the paper.

The referee is correct to point out that the model shown in Fig. 1e does not explicitly show pairing between electrons in the gamma pocket, but in a complete model pairing arises from K, K', and Γ pockets. This primarily affects the upper critical field, so we introduce the concept later in the

manuscript. The strength of Ising protection for each pocket varies over the Fermi surface, as shown schematically in Fig. 4c, with black arrows corresponding to the magnitude of the Ising field along for a few points along the high symmetry line $K' - \Gamma - K$. In our analysis beginning on line 252, we include contributions from all three pockets when computing the k-space average of Δ_{SO} , using the full k-dependence of the valence band splitting, $\Delta_{vb}(\mathbf{k})$. This analysis is crucial to the comparison with MoS_2 since other authors have suggested that the Γ pocket in NbSe_2 and TaS_2 may greatly weaken Ising superconductivity in the latter, p-type materials, whereas we find that reduced Ising protection from the Γ pocket is only a moderate effect. See also the new Fig. S1 in the Supplemental Information, comparing the theoretical upper critical line for singlet pairing including (1) only the K, K' valleys and (2) K, K' , and Γ pockets.

6. Because the Rashba effect is absent in present samples, the model used in MoS_2 is in general not applicable for the present case. The comparison in Fig. S1 needs clearer justification.

We agree that this comparison was unclear and thus we have removed it. Our original goal was to point out that the conventional theory for calculating the upper critical line requires a strong competing effect to agree numerically with experiment. This is now explicitly stated in the revised Discussion, beginning on line 309, and is illustrated in Fig. S1, showing the singlet-pairing theory which vastly overestimates the experimental upper critical field. We now compare this theory to ones including spin-orbit scattering and intervalley scattering, more realistic competing effects for our system (see revised Discussion and new sections in Supplemental Information).

REVIEWERS' COMMENTS:

Reviewer #1 (Remarks to the Author):

The authors have improved substantially the manuscript and all my concerns have been answered satisfactorily.

Therefore, I recommend its publication in Nature Communications.

Reviewer #2 (Remarks to the Author):

The authors considered very carefully all questions raised by the reviewers. Therefore I find that the revised version of their manuscript is suitable for its publication in Nature Communications.